# Performance of Classification Models of Toxins Based on Raman Spectroscopy Using Machine Learning Algorithms

**DOI:** 10.3390/molecules29010197

**Published:** 2023-12-29

**Authors:** Pengjie Zhang, Bing Liu, Xihui Mu, Jiwei Xu, Bin Du, Jiang Wang, Zhiwei Liu, Zhaoyang Tong

**Affiliations:** State Key Laboratory of NBC Protection for Civilian, Beijing 102205, China; zpjbit@163.com (P.Z.); lbfhyjy@sohu.com (B.L.); mxh0511@sohu.com (X.M.); xujw14@mail.ustc.edu.cn (J.X.); dubin51979@163.com (B.D.); roverman@163.com (J.W.); liuzhw07@lzu.edu.cn (Z.L.)

**Keywords:** toxins, Raman fingerprint region, data preprocessing, partial least squares discriminant analysis

## Abstract

Rapid and accurate detection of protein toxins is crucial for public health. The Raman spectra of several protein toxins, such as abrin, ricin, staphylococcal enterotoxin B (SEB), and bungarotoxin (BGT), have been studied. Multivariate scattering correction (MSC), Savitzky–Golay smoothing (SG), and wavelet transform methods (WT) were applied to preprocess Raman spectra. A principal component analysis (PCA) was used to extract spectral features, and the PCA score plots clustered four toxins with two other proteins. The k-means clustering results show that the spectra processed with MSC and MSC-SG methods have the best classification performance. Then, the two data types were classified using partial least squares discriminant analysis (PLS-DA) with an accuracy of 100%. The prediction results of the PCA and PLS-DA and the partial least squares regression model (PLSR) perform well for the fingerprint region spectra. The PLSR model demonstrates excellent classification and regression ability (accuracy = 100%, Rcv = 0.776). Four toxins were correctly classified with interference from two proteins. Classification models based on spectral feature extraction were established. This strategy shows excellent potential in toxin detection and public health protection. These models provide alternative paths for the development of rapid detection devices.

## 1. Introduction

Rapid and precise identification of harmful substances, including toxins and biological agents linked to environmental pollution, is essential for protecting public health [1,2,3]. Detecting targets at early stages is the primary measure for prevention, emphasizing the need to devise efficient detection techniques. Traditional toxin detection methods, such as mass spectrometry, capillary electrophoresis, and biological methods, are both time- and labor-intensive and fail to meet the immediacy and convenience needs of online screening [4,5,6]. Raman spectroscopy is a valuable technique for researching molecular structures. It has been used in various fields, including toxin and virus analysis and environmental monitoring [7,8,9]. The method has numerous benefits, including its convenience, high sensitivity, and compatibility with various types of analytical equipment [10,11,12,13]. Raman spectroscopy can detect targets over long distances, but its application is limited due to weak signals. Enhancing the intensity of spectral signals can convert weak signals into strong signals, enabling sample detection using some nanomaterials. These materials amplify spectral signals to identify a target’s class [14]. Material preparation methods are typically complex and can be time-consuming and labor-intensive when combined with Raman spectroscopy [15]. This joint application does not fully leverage the advantages of Raman spectroscopy. Therefore, developing a novel Raman signal processing method is necessary to classify and detect harmful substances effectively.

Previous research indicates that the combination of Raman spectroscopy techniques and machine learning algorithms facilitates the identification and classification of samples for various applications [16,17,18,19]. The model leverages Raman spectral data’s fingerprint properties and machine learning (ML) algorithms to simplify data processing. The algorithm for feature extraction chooses relevant features for classification from numerous spectral characteristics, and the algorithm for classification labels the main extracted features. The accuracy of classification is heavily reliant on the techniques used in data processing [20,21,22]. Successful classification of toxin samples relies heavily on selecting appropriate methods for data feature preprocessing, extraction, and classification algorithms. The standard approach uses Savitzky–Golay convolutional smoothing (S-G) and multiple scattering correction (MSC) algorithms to achieve noise reduction in spectra [23,24,25]. Machine learning methods are applied for classification tasks, including principal component analysis (PCA), partial least squares discriminant analysis (PLS-DA), and support vector machine (SVM) [26]. For intricate data, deep learning has improved the classification performance of Raman spectroscopy [27,28]. After preprocessing the spectral data, various classification algorithms may produce distinct outcomes. The primary objective is to identify the most suitable combination of these algorithms for the fast detection of toxin samples. The optimal classification performance of the model requires effective collaboration between the preprocessing and classification methods. Different classification algorithms are utilized to assess the data performance processed with various preprocessing algorithms, and the set of models achieving the optimal classification outcomes is selected.

Four toxic compounds and two proteins were utilized to examine the model’s classification performance for harmful substances in the presence of comparable interference. The diversity of data is increased by the various sources of toxins. The classification of toxins with similar structures is worthwhile. Bovine serum albumin has a molecular weight similar to abrin and ricin, while ovalbumin has a slightly lower weight. This comparative study evaluates the efficacy of several Raman spectral preprocessing techniques and classification algorithms. The spectral data underwent preprocessing and were subjected to feature extraction and classification with PCA, K-means, and partial least squares (PLS) algorithms [29,30,31]. PCA and K-means are the most common unsupervised learning methods. PLS-DA is considered the gold standard for performance comparisons [32]. The data and principal components (PCs) were then divided into *k* classes using K-means. This paper presents six sample classes. Therefore, the value of *k* was selected as 6. The preprocessed spectral data underwent PLS-DA and partial least squares regression analysis (PLSR) for classification and regression findings. K-means clustering and PLS classification and regression models were developed, and their robustness was verified using means and validation sets [33,34]. The significance of selecting a spectral feature range for classification results was investigated. This study also examined the feasibility of achieving sample classification with original spectral features and Raman fingerprint regions with relatively concentrated feature areas. The model investigates the capacity for small sample classification and generalization. Up to this point, our research evaluated various preprocessing approaches and classification algorithms. It analyzed the effect of the combination in enhancing the classification and identification performance of toxin Raman spectra. The method is anticipated to facilitate the rapid detection and identification of hazardous samples and boasts significant potential for the classification of toxic substances present in the environment. This study can provide theoretical support for developing new devices for online Raman spectroscopy monitoring in the future. The relevant framework of the proposed classification and prediction model for Raman spectroscopy is shown in Figure 1.

## 2. Results

### 2.1. Raman Spectrum Preprocessing and Peak Assignments

#### 2.1.1. Raman Spectrum Preprocessing

The raw spectral data were preprocessed using MSC, SG, and WT algorithms. MSC was used to correct the baseline shift and offset spectral data using the mean value of spectral data as the ideal spectrum. Finally, the processed data were filtered using SG smoothing (seven-point window, second-order polynomial fitting). WT was used for data wavelet denoising. The wavelet function is Daubechies (db8), and the threshold function is a soft threshold. Similar to the original spectra (OS, 3500–200 cm^−1^) preprocessing, the fingerprint spectra (FS, 1800–400 cm^−1^) are intercepted as new spectra for classification and prediction. The average Raman spectra of the six proteins are shown in Figure 2. The subfigures undergo minimal modifications as a result of the shared source data. The data has undergone minor modifications following pre-processing. All Raman spectra of the six proteins are shown in Appendix A. After being processed with standardization, multiple scattering corrections, and convolution smoothing, the raw data became cleaner (consistent and accurate). Compared with the spectra of several samples, some characteristics are pronounced. These differences are called the features of the Raman spectrum and were widely studied as an essential basis for spectral classification.

#### 2.1.2. Peak Assignments

The corresponding chemical components of the spectral peaks are shown in Table 1. There are 1453 points in each spectrum. Spectral characteristics were observed in protein bands, such as amide I (1640–1680 cm^−1^), amide II (1480–1580 cm^−1^), amide III (1200–1300 cm^−1^), and disulfide bond (490–550 cm^−1^). Other spectral features were found in lipids, CH_2_ scissoring vibration (1420–1450 cm^−1^), and -(CH_2_)_n_- in-phase twist vibration around 1300 cm^−1^. The peak of amino acid in the protein was observed at 570, 575, and 574 cm^−1^ of SEB, abrin, and ricin, respectively, which was not found in BGT or the other samples. The wave number 950 cm^−1^ of SEB was unique in its shift and intensity. The protein peak at 1004 cm^−1^ is a phenylalanine symmetric ring breathing peak [35]. Abrin, SEB, and ricin have peaks at 1209 cm^−1^, representing the tryptophan and phenylalanine C-C_6_H_5_ vibration mode (protein assignment). The wavenumber 1446 cm^−1^ is the CH_2_ bending mode of proteins and lipids found in the spectra of SEB and BGT. The wavenumber 1449 cm^−1^ found in the abrin spectra was confirmed to be a C-H vibration (proteins). The wavenumber 1554 cm^−1^ found in ranges of abrin and BGT belongs to the category of amide II bands. The wavenumber 1552 cm^−1^ in SEB and ricin indicates the existence of a C=C vibration band or tryptophan (protein assignment). It seems that these prominent peaks are quite unique. However, except for SEB, the Raman spectra of BSA, OVA, and the toxins are similar. According to the literature, the protein structures of the three types of toxins are dipeptide chains. The SEB structure is different from theirs [36,37,38,39]. 

### 2.2. Evaluation of Preprocessing Methods Using K-Means and PCA

PCA was applied to evaluate the performance of several preprocessing models. The dimensions of the dataset in this paper were reduced from 1453 to 5. The top two most significant variances were selected to describe the toxins’ spectral properties, accounting for more than 90% of the total variance. According to experience, if the cumulative contribution rate reaches more than 80%, it can be considered that the selected principal components retain most of the information of the original data. K-means was used to cluster preprocessed Raman spectral and PC data. The confusion matrices of the data processed with MSC and MSC-SG are shown in Table 2. The MSC-SG-processed spectral data were correctly classified with K-means algorithm. The classification performance of PC data is not as good as that of spectral data. As demonstrated in Appendix A, the classification results of the raw data and wavelet denoising data are not ideal. The Raman full spectra processed with MSC and MSC-SG were correctly classified with a 100% accuracy rate. The clustering results of the average Raman spectral data also indicate an improvement in the classification performance of the mean data, with an accuracy of 100% (Appendix A).

As shown in Figure 3, the PCA score plots visualize the Raman data processed with MSC and MSC-SG. Figure 3A,B shows the visualization results of the FS and OS spectra processed with MSC. It shows the correct classification of all categories in the OS spectra. Each class was clearly distinguished. Figure 3C,D shows the classification results of the MSC-SG-processed data. The classification results of the FS-MSC-SG data were better than the FS-MSC data. Appendix A shows both spectral data from the FS region. Appendix A shows both spectral data from the OS region. The results of the PCA classification of raw Raman data are shown in Appendix A to investigate the denoising effect of spectral data. Appendix A shows that abrin overlaps with BGT, while BSA overlaps with OVA. The results of the PCA classification of Raman data processed with WT are shown in Appendix A, which indicate that abrin is separated from ricin but overlaps with BGT. The principal component variance contribution rate of OS-MSC-SG data is shown in Appendix A.

### 2.3. Evaluation of Two Raman Spectrum Regions Using PLS-DA

PLS-DA was used to classify the unknown samples with the mixOmics package of R-software (R 4.3.1) on Raman and FTIR spectra. As a supervised method, PLS-DA predicts the sample category by establishing the relationship model between the spectral characteristics and the sample category. Based on previous research results, the data processed with the MSC and MSC-SG methods were used for further analysis. The PLS-DA classification error rate is shown in Appendix A. The solid and dashed lines represent the overall error (overall) and balanced error rate (BER), respectively. The increase in components leads to a decrease in the classification error rate. The error rate reaches its minimum at a component number of 6 and then stabilizes. Therefore, five dimensions are sufficient to achieve optimal classification performance for the model. BER and Mahalanobis distance is sufficient to achieve good performance (with an error rate of 0). The area under the curve (AUC) is 1, and there is a significant difference between different samples (*p* < 0.01, shown in Appendix A). It can be seen that among the three methods (maximum distance, Centroid distance, and Mahalanobis distance) used for measuring vector distance, the Centroid and Mahalanobis distances are more suitable for this model. 

The classification results are shown in Figure 4. As a supervised method, PLS-DA predicts the sample category by establishing a relationship model between the spectral characteristics and the sample category. Figure 4A,C shows the MSC and MSC-SG spectra classification results, respectively. It shows that all classes are correctly classified in FS spectra. Both models show that the spectral classification result of the Raman fingerprint region is better than that of the original area. As a comparison, the classification results of the data before denoising and WT processing are shown in Appendix A.

### 2.4. Evaluation of Two Raman Spectrum Regions Using PLSR

PLSDR algorithms were written in the Python language, and the integrated development environment (IDE) software PyCharm (Community 2021.3) was used. Due to the spectral classification performance processed with MSC and SG, the spectra pretreated with this method will be used in subsequent studies. The ranges preprocessed with MSC and SG were used for regression analysis, with the raw spectra as the control group. The prediction model based on PLSR was used to predict and analyze toxins and interferents. The partial least squares method inputs data as numerical values. The dependent variable, which is the category label, originally has a data type of string. Before conducting regression analysis, it is converted into a numerical data type. In our spectral database, the serial numbers of abrin, BGT, BSA, OVA, ricin, and SEB are 16, 20, 22, 28, 30, and 31, respectively. In contrast to the other two methods, the FS data processed with MSC-SG has the best evaluation results for different spectral data. The relationship between the actual class and the predicted class is shown in Figure 5. The red dot represents the predicted value position of the sample. If the predicted value is consistent with the actual value, then the classification is correct (the dots fall on the green line). The blue line is the fitting line between the actual and predicted values.

Figure 5E shows that the FS-MSC-SG data performed the best—the minimum RMSE results in the suggested number of components as a function of the number of components. In Appendix A, the number of PLS components is highlighted on the plot. The better regression model has a higher number of PLS components, which is 11 in this experiment. The relevant error analysis is shown in Table 3. Cross-validation can solve the problem of insufficient data volume and is crucial for the credibility of predictive models. In contrast, the error results of the FS-MSC-SG model are ideal, with the maximum Rcv value (0.776). Therefore, the data preprocessing methods and reduced Raman spectrum region are validated.

## 3. Discussion

These four protein toxins are very different in structure. The classification outcomes of the data exhibit significant disparities following diverse preprocessing techniques. The failure of wavelet transform denoising may be because WT is very effective for noisy data. However, it is not suitable for spectral data preprocessing in this paper. SG retains the change information of the signal more effectively while filtering and smoothing. The MSC method eliminates spectral differences caused by different scattering levels, thus enhancing the correlation between spectra and data. It is found that some special chemical bonds have a significant contribution to feature extraction. Raman spectral band assignments were studied here. The feature peaks are prominent in the principal component score, corresponding to spectral wavenumber and intensity, such as assignments 510, 853, 1450, and 1664 cm^−1^, which were divided into the S-S band, the tyrosine dimer, CH_2_, and amide I, respectively. This characteristic information of the spectra can be used as the basis for the classification of samples. However, relying on assistance from the human eye to judge these features will reduce efficiency and accuracy, and using computational methods is the best choice.

PCA successfully reduced the dimensions to two dimensions at the cost of losing a small amount of sample information. The cumulative variance contribution rate exceeded 80%, and it is considered that the PCs contain the primary data. The PCs extracted with PCA can be used for classification. The relationship between sample spectra and principal components (PCs) was investigated by integrating K-means clustering with PCA. The results showed that the classification results of the full range were superior to those of the PC analysis. The poor clustering performance of the top ten principal components may be due to insufficient input data. PLS-DA requires that each training sample contains the predicted actual value. PLS-DA discovered the similarities and differences between groups and achieved the classification of different toxin samples. The classification of toxins such as abrin, ricin, SEB, and BGT using PLS-DA has not been reported. The accuracy of the classification models established here for the four toxins and two interfering proteins is 100%. PLS is suitable for small-sample learning. The R-value is 1, indicating that the predicted value equals the actual value. Although the Raman spectral information of proteins often overlaps, this method successfully extracts and classifies feature information. The results show that this model has advantages in quickly organizing spectral features. This strategy is a preferred solution when developing new toxin-detection equipment. These models can potentially be applied in the rapid detection of samples. The running time of each algorithm is less than half a minute (Appendix A). The results of this experiment are expected to provide alternative routes for the innovation of Raman monitoring equipment in the future.

## 4. Materials and Methods

### 4.1. Materials

This work analyzed four kinds of high-purity solid powder toxins and two typical protein samples with spectroscopic instruments. Abrin, ricin, staphylococcal enterotoxin B (SEB), and β-bungarotoxin (BGT) were purchased from Beijing Hapten and Protein Biomedical Institute (Beijing, China). Bovine serum albumin (BSA) and ovalbumin (OVA) were bought as solid powders from Beijing Solarbio Science & Technology Co., Ltd. (Beijing, China) and Shanghai Macklin Biochemical Co., Ltd. (Shanghai, China), respectively. 

The half-lethal dose (LD_50_) of ricin is 3–5 μg/kg (mouse, i. v.). The LD_50_ of abrin is 0.56 μg/kg (mouse, i. v.), which is higher than the toxicity of ricin. The experiment was conducted in an environment with ventilation and emergency management measures for poisoning. The sample did not need to be prepared in advance. 

### 4.2. Raman Spectra Acquisition

The instrument was a confocal Raman spectrometer DXR3 (Thermo Fisher Scientific Inc., Waltham, MA, USA). A 532 nm high-power laser was used in this work with laser wavelength: 532 nm, laser power: 5 mW, objective magnification: 50×, number of gratings: 900 lines/mm, and integration time: 6 s. A small amount of the powder sample was placed on the aluminum sample plate, which was installed in the closed sample chamber underneath the microscope, and the focus was adjusted to make the vision clear. The spectral scanning range was 3500–200 cm^−1^, the resolution was 4 cm^−1^, and the number of exposures was sixty-four. Thirty-six Raman spectra were obtained, including four toxins (N = twenty-four) and two proteins (N = twelve). Subsequently, the results were reported as measured values for preprocessing. A blank background was documented after every five scans. The instrument software (OMINIC Spectra 2.2.0, Waltham, MA, USA) corrected the spectra baselines with a fitting order of 4. Finally, the biological FS and OS data were selected.

### 4.3. Spectra Preprocessing

All spectra were normalized to standard distribution data with a mean of 0 and a variance of 1. Due to noise in the initially acquired Raman spectra, the MSC method was used for baseline correction. The second-order polynomial SG algorithm was used for smoothing and denoising (a window size of 7). We optimized the discrimination ability of the classification model using the different preprocessing methods mentioned above (MSC, MSC-SG, WT). WT is used for analyzing non-static data, such as Raman spectroscopy [51]. WT decomposes the signal into details and approximations [52]. By performing wavelet decomposition on the spectral data by removing noise in the wavelet transform vector, a sample signal can be created using wavelet reconstruction.

### 4.4. Classification and Prediction Methods

K-means is a clustering algorithm that aims to gather similar samples together, simply utilizing the distribution patterns in the sample data itself. The principle of K-means is to initialize k cluster class centers, calculate the distance between the samples and centers, and minimize the distance between similar species and their cluster centers. PCA is an adaptive data analysis technique for reducing the dimensionality of a dataset by creating new artificial variables called principal components (PCs). PCA decomposes the preprocessed data into independent new irrelevant vectors (PCs) [53,54]. Each PC is orthogonal to each other and arranged in order of the percentage of the explanatory variables of the data. A data matrix X can be constructed from vectors obtained from Raman spectroscopy, where each column of this new matrix X contains each sample matrix, and each row represents a variable, i.e., wavenumbers. Finding the first principal component (PC1) is transformed into finding the eigenvector corresponding to the maximum eigenvalue of the X covariance matrix. 

PLS-DA takes into account data dimensionality reduction and regression and classifies and distinguishes regression results. This model can visualize the results and obtain grouped ellipses using calculation. PLS, also known as PLSR, was carried out using the Scikit-learn library. PLSR projects the Raman data onto a subspace of latent variables (LVs), which have maximum covariance with the response(s). The performance results of the machine learning algorithm were obtained using 5-fold cross-validation [55]. 

### 4.5. Performance Evaluation of Models

In the PLS model, the coefficient of determination (R^2^) refers to the sum of the variance the model can explain. The root means square error of calibration (RMSEC), the root mean square of error cross-validation (RMSECV), the correlation coefficient of the cross-validation set (Rcv), the correlation coefficient of the calibration set (Rc), and the factor were considered. When Rcv and Rc values are close to 1, the better the model [56]. Accuracy, confusion matrices, and receiver operating characteristic (ROC) curves are also used to evaluate classification performance [57]. The smaller the RMSE, the more accurate the machine learning model. The RMSE and R were calculated according to the following equations:(1)$$\mathrm{RMSE}=\sqrt{\frac{1}{n}\sum_{i=1}^{n}{\left(c_{i}-{\hat{c}}_{i}\right)}^{2}},$$
(2)$$R=\sqrt{1-\frac{\sum_{i=1}^{n}{\left(c_{i}-{\hat{c}}_{i}\right)}^{2}}{\sum_{i=1}^{n}{\left(c_{i}-\bar{c}\right)}^{2}}},$$
where *n* is the number of samples, $c_{i}$ is the experimental measurement for sample *i*, ${\hat{c}}_{i}$ is the corresponding value obtained with the validation, and $\bar{c}$ is the average value.

## 5. Conclusions

The present study focuses on investigating the preprocessing techniques and classification models used to analyze Raman spectra of protein toxins. Hazardous toxins, such as abrin, ricin, staphylococcal enterotoxin B (SEB), and β-bungarotoxin (BGT), were correctly classified. Some algorithms, including MSC, SG, and WT, were applied for preprocessing Raman spectra, and then PCA, K-means, and PLS were performed to extract spectral features and classify the samples. PCA visualized spectral features by drawing scatter plots using PC scores. The visualization results indicate that the OS spectrum processed with MSC and MSC-SG performs well. The combination model of PCA and K-means verified that MSC-SG is better than the MSC and WT methods, with an accuracy of 100%. The PLS algorithm was used to estimate the classification and prediction performance of the FS and OS spectra. The PLS-DA model completely classifies the fingerprint region datasets of the toxins and proteins. The classification results of the PLSR model show that FS (Rcv = 0.776, RMSECV = 3.488) is superior to OS (Rcv = 0.674, RMSECV = 4.087). The selection of the MSC-SG preprocessing technology improved the classification performance of protein toxins, providing a template for Raman data processing. Combining preprocessing and classification algorithms appears to solve the issue of classifying toxins in the presence of protein interference. This approach is significantly more valuable than solely investigating toxin detection methods. This strategy has good application prospects in toxin identification and public health. The present model can function as an analytical unit for detection instruments and make a significant contribution to the development of novel monitoring devices.

## Figures and Tables

**Figure 1 molecules-29-00197-f001:**
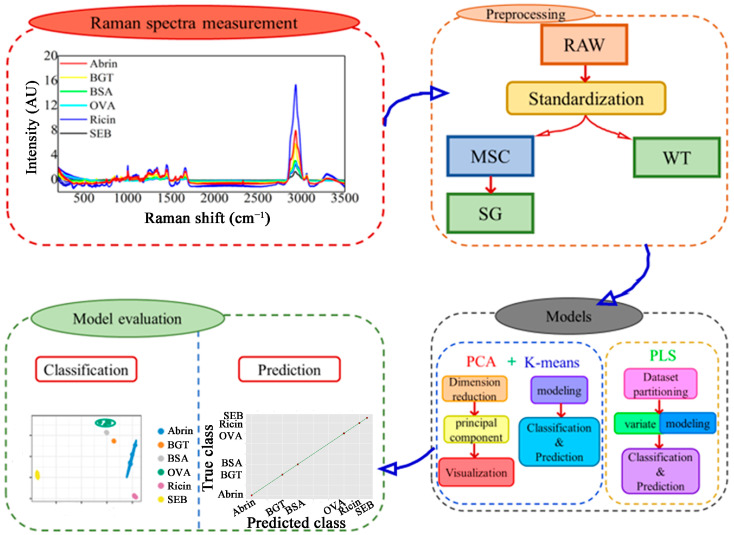
The overall framework of the proposed model. Abbreviations: staphylococcal enterotoxin B (SEB), beta-bungarotoxin (BGT), multivariate scattering correction (MSC), Savitzky–Golay smoothing (SG), wavelet transform methods (WT), principal component analysis (PCA), and partial least squares (PLS).

**Figure 2 molecules-29-00197-f002:**
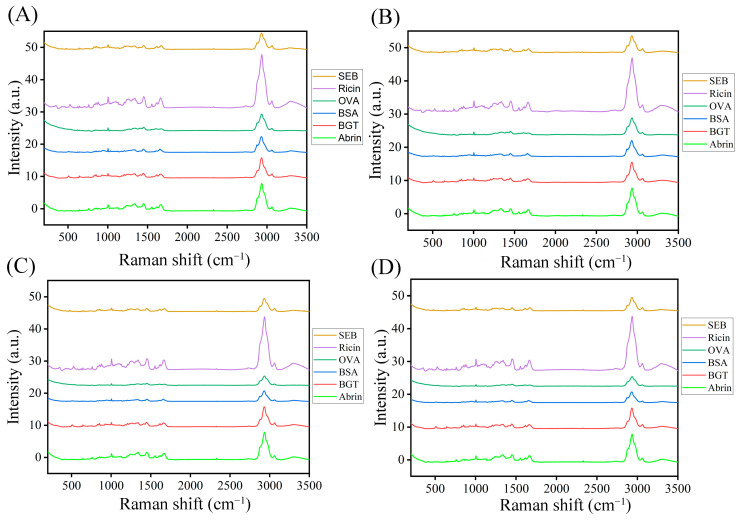
Raman spectra of toxins and proteins processed with raw (**A**), WT (**B**), MSC (**C**), and MSC-SG (**D**).

**Figure 3 molecules-29-00197-f003:**
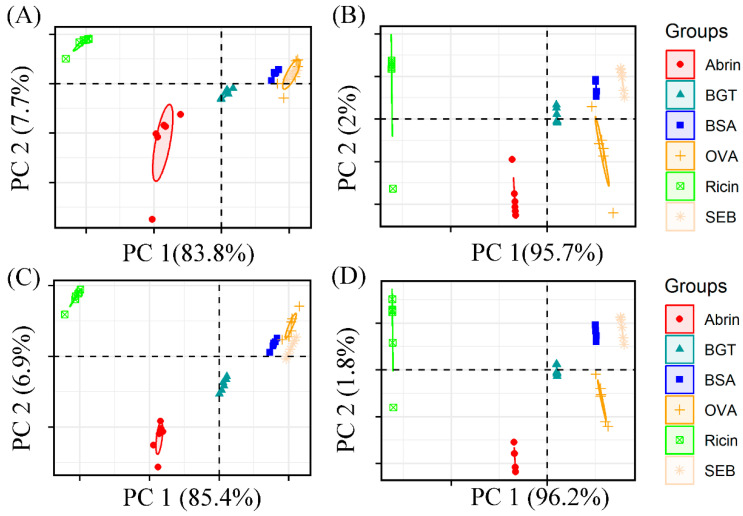
PCA score plots of the fingerprint (**left**) and original region spectra (**right**) were processed with different pretreatments including MSC (**A**,**B**) and MSC-SG (**C**,**D**). Compared with BSA and BGT, the fingerprint region spectrum has better classification performance than the original region.

**Figure 4 molecules-29-00197-f004:**
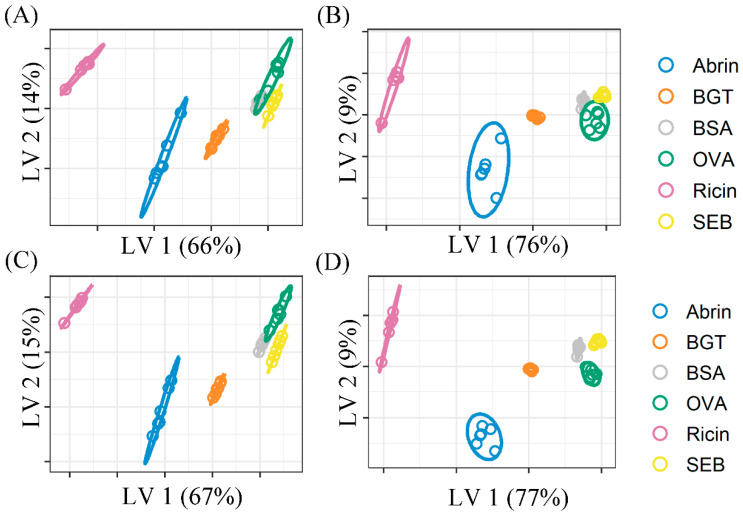
PLS-DA score plots of the fingerprint region (**left**) and the original region (**right**) spectra processed with different pretreatments including MSC (**A**,**B**) and MSC-SG (**C**,**D**). The fingerprint region spectrum has better classification performance than the original light.

**Figure 5 molecules-29-00197-f005:**
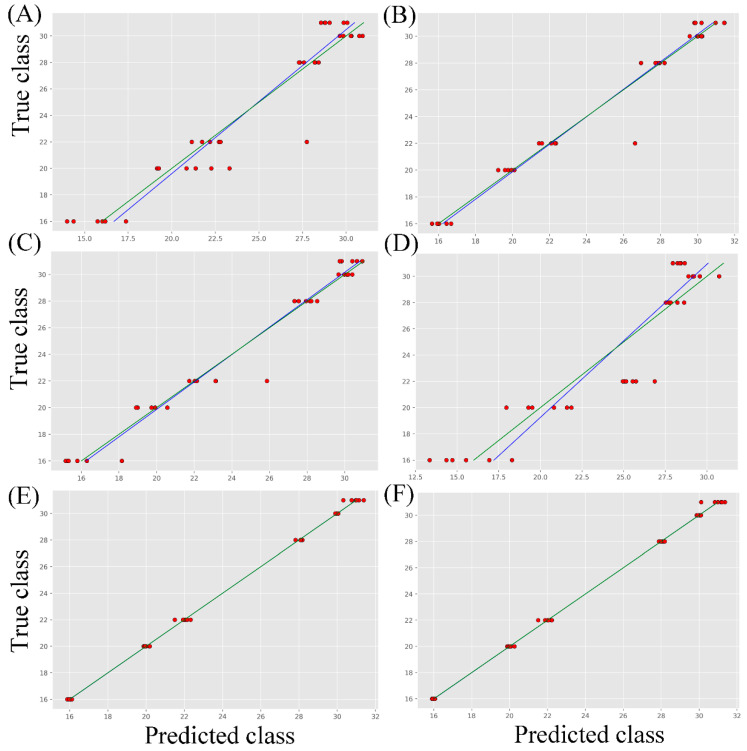
PLSR results of the fingerprint region (**left**) and the original region (**right**) spectra with different pretreatments including raw (**A**,**B**), MSC (**C**,**D**), and MSC-SG (**E**,**F**). The x-coordinate of the point on the green line is equal to its y-coordinate. The blue line represents the regression line that optimally fits the observed and predicted values.

**Table 1 molecules-29-00197-t001:** The peak assignments of Raman bands in the measured samples.

Assignment	Wavenumber (cm^−1^)	This Work (cm^−1^)	References
Tryptophan/cytosine (C), guanine (G)	573	578	[40]
Ring breathing mode of tyrosine	853	853	[40,41]
Phenylalanine, symmetric ring breathing	1002–1005	1004	[42,43]
C-H in-plane phenylalanine (proteins)	1033	1033	[44]
Tryptophan and phenylalanine ν(C-C_6_H_5_)	1209	1209	[44]
Amide III	1230–1300	1231	[45,46]
CH_2_ bending (proteins)	1450	1450	[46,47]
Amide II	1480–1575	1484–1545	[48,49]
Amide I (proteins), C=O stretching (lipids)	1655–1680	1655–1680	[50]

**Table 2 molecules-29-00197-t002:** The confusion matrices of the K-means for Raman spectra and PC data. The data were processed with MSC and MSC-SG methods. The classification results of K-means are expressed in percentages.

Methods	Actual Class	Spectral Data						PC Data					
		* Abrin	* BGT	* BSA	* OVA	* Ricin	* SEB	* Abrin	* BGT	* BSA	* OVA	* Ricin	* SEB
MSC	Abrin	100						100					
	BGT		100						100				
	BSA			100						100			
	OVA				83.3						66.7		
	Ricin					100						83.3	
	SEB						100						0
MSC-SG	Abrin	100						83.3					
	BGT		100						100				
	BSA			100						100			
	OVA				100						83.3		
	Ricin					100						83.3	
	SEB						100						50

* Abrin: The predicted class of abrin. * BGT: The predicted class of BGT. * BSA: The predicted class of BSA. * OVA: The predicted class of OVA. * Ricin: The predicted class of ricin. * SEB: The predicted class of SEB.

**Table 3 molecules-29-00197-t003:** Performance of the PLS model with different processing spectra.

Model	RMSEC	RMSECV	Rc	Rcv
FS—Raw	1.567	4.271	0.959	0.636
FS—MSC	0.938	4.256	0.985	0.639
FS—MSC—SG	0.192	3.488	0.999	0.776
OS—Raw	0.897	4.602	0.987	0.555
OS—MSC	2.087	5.087	0.926	0.392
OS—MSC—SG	0.212	4.087	0.999	0.674

FS, fingerprint region spectra; OS, original region spectra.

## Data Availability

The data presented in this study are available in article and Appendix A.

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
