# Peer review of "Performance of Classification Models of Toxins Based on Raman Spectroscopy Using Machine Learning Algorithms"

_molecules, 2023, doi:10.3390/molecules29010197_

Round 1

Reviewer 1 Report

Comments and Suggestions for Authors

The manuscript describes the application of Raman spectroscopy and machine learning to detect four toxins, namely, abrin, ricin, staphylococcal enterotoxin B (SEB), and bungarotoxin (BGT).

Rapid and accurate identification of toxins is an important task. Raman spectroscopy method can be very useful in this area due to the speed of obtaining information. The authors provide a comparative study of proteins that may interfere with the detection of toxins. The authors use well-known machine learning techniques and demonstrate excellent separation of toxins from interfering proteins.

I consider the manuscript topic is original and relevant in the field. This comparative study evaluates the efficacy of several Raman spectral preprocessing techniques and classification algorithms. This work may be useful for Raman spectroscopists. The authors demonstrate that machine learning methods can improve Raman separation of protein toxins without the use of additional methods such as SERS.

Let's look at parts of the manuscript in more detail.

1.Introduction

The authors provide brief data on the dangers of toxins and write that methods for detecting them are “both time and labor-intensive and fail to meet the immediacy and convenience needs of online screening”. A more detailed description of traditional toxin detection methods would be worthwhile; moreover, many general reviews can be found, e.g. “Detection Methodologies for

Pathogen and Toxins: A Review”, Sensors 2017, 17, 1885; doi:10.3390/s17081885.

The authors then cite a number of articles on SEPS, e.g. [4], [5], [8], [11], [12] and etc. The authors write: ”Material preparation methods are typically complex and can be time-consuming and labor-intensive when combined with Raman spectroscopy”. However, SERS can significantly improve the detection limit of toxins compared to Raman. I think the authors should provide comparative characteristics of SEPC and Raman for detecting toxins.

The authors chose 4 protein toxins for analysis, but did not justify their choice. Why were these particular protein toxins studied? It would be useful to indicate which analytical methods are used to detect these specific protein toxins. Has Raman spectroscopy been used previously to analyze these toxins?

To extract informative features in Raman spectra, the Support vector machine (SVM), Random forest (RF), and XGboost methods are also used (see doi: 10.3390/pharmaceutics15010203).

There are other methods to improve classification accuracy of Raman spectroscopy

(https://doi.org/10.1021/acs.jcim.3c00761; https://doi.org/10.3390/analytica3030020). These machine learning methods should have been mentioned in the introduction. Why don't authors use these methods in their work?

2. Results

It would be useful to provide data on the structure of the protein toxins being studied. This would help explain the differences in the Raman spectra. It is necessary to compare the Raman spectra obtained in the work with known data. The authors provide a comparison for ricin only (see [33]).

The authors studied pure preparations of protein toxins and proteins. It would be good to study a mixture of these proteins. Will there be good differentiation of toxin mixture by the machine learning methods used?

The caption to Figure 1 should also contain a description of the abbreviations. The abbreviations “DWT”, “FS”, “OS” should be deciphered upon first mentioned in the text. I did not find reference [28] in the text of the manuscript.

The Discussion and Conclusion are quite consistent with the evidence and arguments presented and they address the main question posed.

The references are appropriate. The references have been used adequately, mainly over the last 5 years. I believe that the list of references will be expanded upon responses to the comments above.

The manuscript may be accepted after responding to my questions and comments.

Reviewer 2 Report

Comments and Suggestions for Authors

The manuscript "Performance of Classification Models of Toxins Based on Ra- man Spectroscopy Using Machine Learning Algorithms" describes different methodologies for discrimination and classification of Raman spectra of different proteins and deserves a careful consideration. Hovewer some section, images, and sentences need to be clarified and improved. So I recommend minor revisions as in the detailed list that follows:

1) Page3 lines 97-100: "The average Raman spectra of the six proteins are shown in Figure 2. The original region Raman spectra are shown in Figure S1. After being processed by standardization, multiple scattering corrections, and convolution smoothing, the raw data becomes cleaner (consistent and accurate)...."

Looking at both Figure 2 and Figure S1 the main peak of SEB at about 2900 cm-1 results in a strong intensity reduction after the first step in figure 2 (RAW -> WT) and after the second step in figure S1 (WT -> MSC), that is not well explained in the text and it is not observed in the spectra of other proteins. 

I invite the authors to comment on both these points and add an explanation in the manuscript

2) Page 5 line 139: "Figure 3A and 3B show the visualization results of the FS and OS..." the acronyms "FS" and "OS" hadn't been previously specified in the text. I guess they refer to "Fingerprint Spectra" and "Original Spectra" but it is not specified. Please add. Moreover, the FS interval taken into account is not reported in any part of the text. Please add in this point of the manuscript.  

3) Page 5 line 141-142: "Figure 3C shows the classification results of FS-MSC-SG data. The classification results of OS spectra were better." These two sentences seems to me to have any sense. Do you mean: "Figure 3C shows the classification results of FS-MSC-SG data. The classification results of FS-MSC-SG data were better than FS-MSC?" If so please correct.

4) Page 5 lines 142-147: "The results of PCA classification of raw Raman data are shown in Figure S2A and S2B..." Figure S2A and S2C and Figure S2B and S2D are exactly the same. The points in S2A and S2C are in the exact position, as in S2B and S2D. The only differences in these couples are the PC-scores. That is impossible given the data reported in Figure S1 A-D. Please put the correct data in Figure S2 A-D  

5) Page 6 Table 2: it seems that the MSC-PC-data of SEB are never correctly classified by k-means, that is quite strange. Therefore, in which class k-means classify MSC-PC-SEB data? I would invite the authors to report and comment this datum in the text.

6) page 9-10 lines 266-267: "34 Raman spectra were obtained, including four toxins (N = twenty-five) and two proteins (N = nine)"

Please add here the exact number of spectra for each compound as read from your PLSR results of figure 5:

6 points for number 16 (abrin)

6 points for number 20 (BGT)

6 points for number 22 (BSA)

6 points for number 28 (OVA)

5 points for number 30 (ricin)

5 points for number 31 (SEB)

This means that the reported number of spectra for each class of substances on page 10, line 267 (25 spectra for toxins and 9 spectra for proteins) is wrong and should be corrected in 22 spectra for toxins (6 for abrin, 6 for ricin, 5 for SEB, 5 for BGT) and 12 spectra for proteins (6 for BSA and 6 for OVA). If not the results of figure 5 are wrong. So I invite the authors to clarify in this question and in the text as described. 

Comments on the Quality of English Language

Minor editing of English language should be considered.

Round 2

Reviewer 2 Report

Comments and Suggestions for Authors

Accept. The manuscript has been significantly improved.